# Inflammation of the Pleural Cavity: A Review on Pathogenesis, Diagnosis and Implications in Tumor Pathophysiology

**DOI:** 10.3390/cancers14061415

**Published:** 2022-03-10

**Authors:** Georgia Karpathiou, Michel Péoc’h, Anand Sundaralingam, Najib Rahman, Marios E. Froudarakis

**Affiliations:** 1Pathology Department, University Hospital of Saint-Etienne, 42055 Saint-Etienne, France; michel.peoch@chu-st-etienne.fr; 2Oxford Centre for Respiratory Medicine, Nuffield Department of Medicine, University of Oxford, Oxford OX3 7LE, UK; anand.sundaralingam@ouh.nhs.uk (A.S.); najib.rahman@ndm.ox.ac.uk (N.R.); 3Pneumonology and Thoracic Oncology Department, University Hospital of Saint-Etienne, 42055 Saint-Etienne, France; marios.froudarakis@chu-st-etienne.fr

**Keywords:** pleura, non-specific pleuritis, pleural effusion, mesothelioma in situ, pleural pathology, mesothelial

## Abstract

**Simple Summary:**

The pleura is a cavity whose pathology ranges from simple fluid accumulation to tumor development, all inducing important consequences in patents health, and usually having an important association with local inflammation. Understanding the pathophysiology of pleural inflammation helps the development of the correct treatment strategies and opens new windows in pleural research. Thus, the aim of this review is to present the etiologies and the pathophysiological mechanisms of pleural inflammation with a special interest in their role on tumor development and diagnosis.

**Abstract:**

Pleural effusions are a common respiratory condition with many etiologies. Nonmalignant etiologies explain most pleural effusions and despite being nonmalignant, they can be associated with poor survival; thus, it is important to understand their pathophysiology. Furthermore, diagnosing a benign pleural pathology always harbors the uncertainty of a false-negative diagnosis for physicians and pathologists, especially for the group of non-specific pleuritis. This review aims to present the role of the inflammation in the development of benign pleural effusions, with a special interest in their pathophysiology and their association with malignancy.

## 1. Introduction

Pleural effusions are a common respiratory condition with 1.5 million new cases diagnosed per year in the United States [1]. Of those, 500,000 are attributed to congestive heart failure and 150,000 to malignancy [1]. Almost 70% of pleural effusions will be diagnosed based on clinical history, physical examination and pleural fluid analysis, the rest warranting further work-up, to prove that they are tuberculosis (TB)-associated in TB endemic areas, but likely malignant in low-TB-prevalence areas [2].

Malignant pleural effusions are most commonly of metastatic origin, usually from lung and breast primaries, with mesothelioma accounting for about 12% of the cases [3,4]. Malignant pleural effusion is associated with poor prognosis showing a median overall survival of 11 months depending on histological type and performance status [4]. Furthermore, even in this advanced tumoral setting, the inflammatory nature of malignant effusions significantly impacts on prognosis [3,5], whereas achieving pleurodesis in malignant pleural effusion also imparts a survival benefit in these patients [6], but through unknown mechanisms. On the other hand, non-malignant etiologies explain most pleural effusions and understanding the key events of benign pleural pathologies is important to better treat these patients [7]. The annual hospitalization rate showed a significant decrease in malignant pleural effusion and pleural tuberculosis, but an increase for nonmalignant pleural effusion in the United States, with empyema showing the higher costs and length of stay [8]. Despite nonmalignant, benign pleural effusions can be associated with poor patients’ survival. In a large prospective study of nonmalignant pleural effusion patients, the one-year mortality rate for patients with cardiac, renal and hepatic failure were 50, 46, and 25%, respectively, with bilateral and transudative effusions being associated with worse prognosis [9]. Furthermore, diagnosing a benign pleural pathology always harbors the uncertainty of a false-negative diagnosis for physicians and pathologists. This review aims to present the role of the inflammation in the development of benign pleural effusions, with a special interest in their pathophysiology and their association with malignancy.

## 2. Pleura Definition: Anatomy, Histology, Physiology

The pleura (from the Greek word for side) is a mesothelial membrane underlying the thoracic cage; it forms a closed space, the pleural cavity, a virtual space containing only a few ml of fluid—0.26 mL/kg of body weight [10]—between the visceral (covering the lung) and parietal (covering the thoracic wall, the diaphragm and the mediastinum)—membrane (Figure 1). There are two separate individual pleural cavities. Having said that, one should keep in mind that fenestrations in the mediastinum resulting in interpleural communications between the right and left pleural cavities have been described in several species, and may present a potential risk for humans too, since it could result in a bilateral pneumothorax (the buffalo chest named after an anecdote which stated that the Native Americans could kill a North American bison with a single arrow in the chest) [11].

The exact function of the pleural space is not clear, especially when considering that there are animals, such as the elephants, that do not have a pleural space, and that after pleurodesis, an iatrogenic obliteration of the pleural space, respiratory function is also maintained. For physicians, its importance lies in the formation of pleural effusions. The normal pleural fluid production and absorption depends on the hydrostatic and oncotic pressure gradients between the parietal and visceral pleura and the pleural space and is principally mediated by the parietal pleura since it has higher hydrostatic pressures (fluid production) and contains the lymphatic stomata (fluid resorption) [10]. The most common cause of increased pleural fluid is increased interstitial edema which is the predominant mechanism for the formation of parapneumonic effusions (see below), congestive heart failure, pulmonary embolism and acute respiratory distress syndrome [12]. When the pleura becomes inflamed, capillary permeability certainly contributes to pleural fluid formation, whereas lymphatic obstruction commonly contributes to malignant pleural effusions [12].

As for the clinical impact of a pleural effusion, the most common presentation is breathlessness, which can significantly impair the quality of life; the mechanisms behind this manifestation and the relief following thoracentesis are more complicated than a simple explanation of a compressed lung parenchyma, and it seems that the expansion of the thoracic cage and especially the displacement of the diaphragm with the resulting loss of its efficiency is the most probable cause of breathlessness [13].

Histologically, the pleura is lined by the mesothelium, an epithelium derived from the mesoderm. It is formed by a single layer of cells but is underlined by its stroma containing fibroblasts, endothelial cells, immune cells and extracellular matrix (Figure 2).

The visceral pleura (Figure 3) contains the following order of layers: a very thin layer of mesothelial cells, an underlying fibro-collagenous tissue, an external elastic lamina, a loose connective tissue which contains vessels, nerve fibers, and an internal elastic lamina; the latter two layers are continuous with the underlying interlobular connective tissue and the elastic layer of the alveoli, which allows for the distribution of mechanical forces during respiration [14,15].

The parietal pleura has a similar structure and lies on a layer of adipose tissue; this in turn reposes on a fibrocollagenous tissue attached to the thoracic cage. This extrapleural layer of connective tissue provides a surgical plane to separate the parietal pleura from the thoracic wall [14]. Blood in the parietal pleura is supplied through intercostal, internal thoracic and musculophrenic arteries, and its venous drainage also occurs through the intercostal veins, while blood supply and drainage of the visceral pleura occurs through bronchial and pulmonary vessels [14,15]. Lymphatics of the visceral pleura drain through pulmonary lymphatics to the hilar lymph nodes, whereas the anterior parietal/diaphragmatic lymphatics drain to the internal mammary chain, and the posterior parietal/diaphragmatic lymphatics drain to intercostal, paravertebral and occasionally to retroperitoneal nodes [16]. One characteristic structure of the parietal lymphatics is the “stoma” (from the Greek word for mouth), a lymphatic channel of up to 12 μm in diameter which directly opens to the pleural space and drain into the subpleural lymphatic system [15]. The parietal pleura receives a rich innervation via the intercostal nerves [15].

## 3. Pleural Effusion Etiology

Classically, pleural effusions are classified as exudates or transudates and are first characterized by any of the following: pleural fluid protein/serum protein ratio > 0.5, pleural fluid LDH/serum LDH ratio > 0.6, or pleural fluid LDH > 200 IU/L [10]. In the etiologies of exudates, one can find malignancies, paramalignant conditions (pleura effusions reactive to underlying lung cancer, atelectasis, and radiation), parapneumonic pleural effusion (reactive due to underlying pneumonia), infections (TB, empyema, parasitic, viral), cardiac injury (post-coronary artery bypass surgery), pulmonary embolism, collagen vascular disease (systemic lupus erythematosus, rheumatoid arthritis), abdominal diseases, such as pancreatitis, or gynecologic diseases (such as the Meigs syndrome), sarcoidosis, and drug-induced pleural effusions. In the transudates category, etiologies include congestive heart failure, hepatic and peritoneal dialysis-related hydrothorax, nephrotic syndrome, hypoalbunaimia, chylothorax, again pulmonary embolism, atelectasis, sarcoidosis, and other diseases, such as yellow nail syndrome, amyloidosis, asbestos exposure [1,10,17]. Some important points of the following section are summarized in Table 1.

The incidence of tuberculous pleural effusion (Figure 4 and Figure 5) in tuberculosis patients varies from 3 to 25% and thoracoscopy shows 92 to 100% sensitivity in diagnosing tuberculous pleural effusion [17].

The incidence of pleural infection is 6.7–9.9 cases/100,000 population [18]. Pleural infection shows an overall mortality of 10% at 3 months and 19% at 12 months [19]. Pleural space infection despite being known since thousands of years, harbors various terminologies that could lead to some confusion [18]. Parapneumonic effusion is defined as any pleural effusion secondary to viral or bacterial pneumonia or lung abscess [12]. This parapneumonic effusion can be complicated when a pleural infection is present (and often requires an invasive procedure, such as tube thoracostomy, to resolve [12]) which separates it from uninfected parapneumonic effusions occurring in almost 20–50% of patients with community-acquired pneumonia; primary pleural infections occurring without contiguous lung infection are seen in 56 and 73% of community-acquired and hospital-acquired pleural infections, respectively [18]. The term empyema refers to the presence of pus of bacteria in the pleural space [12]. As previously described, there is an initial exudative phase resulting from proinflammatory mediators, followed by a fibrinopurulent phase with invasion of bacteria in the pleural space, loculation and septations, and finally an organization phase, with fibroblast proliferation and pleural thickening [18]. Pleural space is more hypoxic in comparison to the lung, and this is suggested to explain the rare infection of the pleura by atypical bacteria, such as Mycoplasma and Legionella species, despite they are common causes of pneumonia [18]. *Staphylococcus aureus*, *viridans*-group *Streptococci*, and *Streptococcus pneumoniae*, in descending order of frequency followed by Enterobacteriaceae and Pseudomonas species, are the principal pleural infections [18]. 

It is, however, worth reiterating that pleural fluid culture is positive in only about half of the cases [20]. One intriguing question is if the indwelling pleural catheter which can be used for the management of recurrent (mainly malignant) effusions can be complicated by pleural infection; the reported frequency varies from 2 to 12% and they are predominantly represented by *Staphylococcus aureus* and gram-negative organisms [21]. Interestingly, the survival of patients with malignant pleural effusion managed with indwelling pleural catheters finally being infected (3.7%) may be improved after this infection [22], further highlighting the probable role of the inflammation in patients’ prognosis. Fungi are uncommon in pleural infections representing 3% of cases, mostly corresponding to *Candida* species [18]. Pleural effusion as a manifestation of cryptococcosis is extremely rare but should be suspected in immunocompromised patients; since pleural effusion cultures for *Cryptococcus* are usually negative, probably due to the small number of fungi in the pleural fluid, pathologists should keep this suspicion high when examining immunocompromised patients’ pleural biopsies to avoid overlooking it, and colorations such as Methemenamine Silver and PAS could help in the diagnosis [23]. It is postulated that the release of antigens, rather than the fungus itself, is responsible for pleural effusion, thus, cryptococcal antigens can be detected in the pleural fluid, similarly to the blood and cerebrospinal fluid [23]. Viral pneumonias can be also associated with pleural effusions with non-COVID-19 pneumonias showing pleural effusion in 25% of the cases compared to 3% of COVID-19 pneumonias [24], attributed often to comorbidities or directly to the viral infection [25]. It also seems that the presence of pleural effusion in COVID-19 patients is associated with disease severity and higher mortality [26]. In extremely rare situations, myiasis, a parasitic infestation by dipterous larvae of flies, typically occurring in tropical and subtropical regions mostly in cutaneous or head and neck localizations, has been reported in the pleural space, probably inserting through the thoracostomy tube site in a pleural angiosarcoma patient [27].

Heart failure provokes increased hydrostatic pressure in pulmonary vasculature leading to fluid leak into the pleural space; almost 90% of patients with decompensated heart failure have pleural effusions [28]. In addition, pleural effusions are found in almost all patients immediately after coronary artery bypass grafting, but most resolve over time [28]. Almost 7% of patients with chronic renal failure and 20% of patients treated with long-term hemodialysis for renal failure develop pleural effusion; of those, 60% are associated to hypervolemia, the rest associated with heart failure, uremia, infection [28].

Inflammatory bowel disease can rarely cause, directly (associated to the systemic inflammatory nature of these diseases and being usually exudative and neutrophilic predominant) or indirectly (associated to medications, infections due to immune suppression, or subdiaphragmatic disease activity and complications) a pleural effusion [29]. Hepatic hydrothorax, defined as a pleural effusion in a patient without evidence of other cardiopulmonary disorders, is reported in 5 to 15% of patients with cirrhosis and portal hypertension, and is associated with liver failure and sodium/water imparlance [30]. Acute pancreatitis patients present with pleural effusion in almost half of the cases, of which 72% are bilateral, 23.7% left-sided and 3.9% right-sided, with probable mechanisms contributing to the formation of pleural effusion in these cases being the transdiaphragmatic lymphatic blockage, a pancreaticopleural fistula formation due to leakage of pancreatic enzymes, or exudation of fluid into the pleural cavity from the subpleural diaphragmatic vessels [31].

Connective tissue diseases can be associated with pleural involvement, which varies from the formation of pleural effusion to simply pleural inflammation/thickening, thus presenting with various manifestations ranging from dyspnea to pleuritic chest pain and fever [32]. Incidence of pleural effusion is reported in 5–20% of rheumatoid arthritis patients, 17–60% of systemic lupus erythematosus patients (SLE), in 5–55% of granulomatosis with polyangiitis patients and in 7% of systemic sclerosis (scleroderma) patients [32]. In a large study of the Chinese SLE Treatment and Research group database, it was shown that 10% of patients with systemic lupus erythematosus will develop pleural effusion (16% will develop serositis, meaning pleural or/and pericardial involvement) and serositis is more frequently found in patients with nephropathy, interstitial lung disease, pulmonary arterial hypertension and hematologic involvement [33]. The serosal involvement in SLE is considered to be a direct reflection of the SLE activity, since hypocomplementemia and elevated anti-dsDNA antibody levels—both factors of active SLE disease—are independent risk factors for serositis in SLE patients, thus SLE-related serositis is thought to be caused by immune complex activation and direct binding of anti-dsDNA antibodies to the mesothelium [33].

Immunoglobulin G4-related disease (IgG4-RD) is a fibroinflammatory disease with systemic manifestations with an estimated incidence in Japan from 0.258 to 1.08/100,000, showing male predominance and an onset after 40 years [34]. Its pathogenesis remains unclear, but a type-2 helper cell (Th2) immune response is highly implicated [34]. Thoracic involvement occurs in almost half of the patients [34]. Similarly, in a UK-based cohort of 53 IgG4-RD patients almost half showed thoracic involvement, with mediastinal lymphadenopathy being the most common, but a variation of clinical manifestations, such as interstitial lung disease, pleural thickening and effusion being described [35]. Patients with thoracic involvement also had higher levels of serum IgG4 [35]. Pleural IgG4-RD presents as nodular or diffuse pleural thickening and pleural effusion with or without underlying parenchymal disease [34]. Histopathological sections of IgG4-RD show storiform fibrosis, dense lymphoplasmocytic infiltrate, obliterative phlebitis, and high IgG4+ plasma cells with the proposed cut-offs varying with the site of involvement; in the lung and pleura, it is suggested to be at >50 IgG4+ plasma cells/high power field for surgical biopsies and >20 IgG4+ plasma cells/high power field for nonsurgical biopsies, and an IgG4+/IgG+ ratio > 40% [34].

Pleural effusions in systemic amyloidosis are considered rare and mostly associated with primary systemic amyloidosis (AL) rather than secondary (AA) amyloidosis [36]. In a study [37] of 636 AL patients referred to the Boston University Medical Center between 1994 and 2001, 35 (6%) had large and refractory pleural effusions, and by comparing them to patients with cardiac amyloidosis without pleural effusion, the authors supported a non-cardiogenic mechanism of pleural disease, with amyloid infiltration of the pleura being proposed as the principal mechanism for these effusions (Figure 6 and Figure 7), while cardiopathy is suggested to contribute to, but not being sufficient for creating these effusions [36]. A more recent study [38] from the Medical University of Vienna including 143 patients with cardiac AL or transthyretin (ATTR) amyloidosis registered between 2012 and 2019 showed isolated pleural effusion in 35, isolated pericardial effusion in 24, and both pericardial and pleural effusion in 19 patients, thus revealing more frequent pleural effusions than previously suggested. The presence of pleural effusion in this study did correlate with right ventricular function in AL patients and with poorer prognosis [38]. The authors also showed that patients with ATTR and pleural effusion had lower albumin levels suggesting that in this form of cardiac amyloidosis, the pathophysiologic mechanism probably is different, associated with decreased plasma oncotic pressure [38].

The yellow nail syndrome, a likely acquired condition of unknown pathophysiology is characterized by yellow slow-growing nails, lymphedema and chronic respiratory manifestations, such as pleural effusion, bronchiectasis, chronic sinusitis, and recurrent pneumonias [39]. Despite, the suggested mechanism of disease is a probable lymphatic dysfunction, pleural biopsies in these cases reveal no specific findings such as fibrosis and chronic inflammation [39].

The treatment of nonmalignant noninfective pleural effusion targets mainly the underlying cause; in some patients, direct intervention to remove or prevent fluid accumulation is needed [28]. This mainly includes pleural fluid aspiration, indwelling pleural catheters and pleurodesis [28]. For infectious causes, antibiotics is the most important initial step in their treatment, while intrapleural fibrinolytic therapy, chest drain, thoracoscopy and surgery can be also used [40].

## 4. Pathophysiology of Pleural Inflammation

A simplified view of the pleural space inflammation starts by the injury of the mesothelial cells which will induce the influx of neutrophils, the increase in vascular permeability and the activation of the coagulation/fibrosis cascade. Neutrophils accompany various pleural diseases; they are abundant in complicated parapneumonic effusion and empyema and in abdominal causes such as pancreatitis and abscesses [41]. In response to bacterial endotoxin and cytokines, interleukin 8 (IL-8) is considered the principal chemotactic factor recruiting neutrophils (Figure 8, Figure 9, Figure 10 and Figure 11) in the pleural space [41].

IL-8 has been shown to be produced in rabbit pleural space and by mesothelial cells in vitro but also in the pleural fluid of patients with empyema where its levels is being correlated to the pleural neutrophil count and the levels of tumor necrosis factor-alpha (TNFa) [42]. IL-8 is mainly produced by mesothelial cells, but immune cells such as macrophages and lymphocytes could contribute to its levels [42]. Then, both bacterial cell wall components and inflammatory mediators will increase the vascular permeability [41]. Furthermore, they will induce the expression of tissue factor on mesothelial and endothelial cells initiating the coagulation cascade [41]. Since all these products cannot be easily drained away from the pleural cavity, compared with, for example, what happens in pneumonia, the inflammation can prevail and if therapeutic intervention in the form of drainage is not initiated, deposition of fibrin and proliferation of fibroblasts will prevail [41].

At initial stages of “simple” parapneumonic effusion (without pleural infection), the effusion is thought to arise by transfer of interstitial fluid of the underlying inflamed lung parenchyma through the visceral pleura; pleural inflammation alone in these stages is insufficient to result in significant effusion, as suggested by the frequent pleuritic chest pain in patients with pneumonia, but only the minority of them will have detectable effusion [7]. Then, pleural inflammation with neutrophil migration and cytokines accumulation will lead to intercellular “gaps” between mesothelial cells further enhancing permeability, which along with vascular permeability leads to additional fluid accumulation [7].

At this stage, if the underlying pneumonia is treated by antibiotics, the effusion may resolve [7]. If lung parenchyma inflammation/infection persists, bacterial invasion of the pleura can occur, provoking the depression of the fibrinolytic capacity of the pleural cavity, leading to fibrin deposition and eventually pleural adhesions [7], although the precise mechanism whereby bacteria invade the pleural space is not understood. This can progress in pleural scarring with entrapment of the lung, and at this stage lysis of the collagenous fibrous tissue will be difficult by other means such as fibrinolytic agents, requiring surgical intervention [7].

Neutrophil extracellular traps (NETs) are web-like structures formed by a scaffold of chromatin containing cytotoxic molecules released by neutrophils; they are defense mechanism since they trap and kill microorganism, but they are also implicated in various diseases, including cancer [43]. Infection of the pleural space is characterized by excessive accumulation of fluid and inflammatory cells, and only recently, NETs have been identified as an important component of this effusion compared to other effusions, such as malignant or transudative ones; this could explain the efficacy of DNase in the treatment of empyema, which reduces the viscosity of the pleural fluid [44].

### 4.1. Immune Cells in Pleural Effusion

Lymphocytes are usually associated with malignant disease or tuberculosis [41]. They are often CD4+ T-helper cells. In these cases, the chemotactic factors involved are lymphocytic chemotactic factor (LCF), monocyte chemotactic peptide (MCP-1) and IL-8 [41]. In tuberculous etiology, the pleural effusion usually develops after rupture of subpleural granulomas, and a delayed hypersensitivity reaction will ensue in the pleural space with a principal role of IFNγ in its development [41]. MCP-1 is a proinflammatory cytokine produced by mesothelial cells, endothelial cells and inflammatory cells and promotes migration and infiltration by monocytes, T-lymphocytes and natural killer cells. Its role in pleural effusion formation has been shown also in a cancer model [45] and in a model of acute pleural inflammation [46].

Apart from the macrophages than can migrate into the pleural space, the serosal cavities contain a distinct population of cavity-resident macrophages dependent on WT1 expression by mesothelial cells and underlying fibroblasts to maintain their GATA6 expression, responsible for the serosal localization of these macrophages and their function [47]. The immunosuppressive environment of peritoneal and pleural cavities which are frequent sites of cancer progression could be related to these cavity-resident macrophages which express Tim-4 in comparison to tumor-associated macrophages; this expression results in sequestration of cytotoxic T-cells by Tim-4+ cavity-resident macrophages [48]. Interestingly, it is suggested that cavity-resident macrophages do not invade underlying lung, nor do they play a role in tissue repair after injury [49].

Eosinophilic pleural effusion is characterized by at least 10% eosinophils in the pleural fluid. Eosinophils in pleural effusions are usually associated with pneumothorax, asbestosis, pulmonary infarction, sarcoidosis, collagen tissue disease, drug reactions, and parasitic and fungal infections, or will be idiopathic [41,50]. Pneumothorax-associated eosinophilia of the pleural fluid (Figure 12) is considered a type-2 innate immune response characterized by the production of IL-33 through a Th2- independent mechanism [51].

### 4.2. Cytokines in Pleural Effusion Formation

Regarding other molecules participating in pleural effusion, matrix metalloproteinases (MMP) and their antagonists, the tissue inhibitors of metalloproteinases (TIMP) are present in pleural effusions, with some of these molecules, such as MMP-1, MMP-2 and TIMP-1 being constitutively present in the pleural fluid, while MMP-9 and TIMP-2 are found in specific diseases states [52]. A study showed that all MMP-8, MMP-9, TIMP-1, IL-6, VEGF, and TGF-b1 are higher in exudates compared to transudates and that MMP-8 and IL-6 are significantly higher in tuberculosis patients than in cancer patients [53]. In addition, mesothelial cells are an important source of VEGF which is known to provoke vessel permeability and thus pleural effusion; by using in vitro and in vivo models, one of the factors shown to induce VEGF production by mesothelial cells is the transforming growth factor β (TGF-β) [54]. Notably, VEGF levels in pleural fluid is higher in patients with residual pleural thickening after pleural infection [55]. In another animal model of acute pleural inflammation (induced by talc or silver nitrate), the use of an anti-VEGF antibody revealed significant reduction of pleural fluid volume, IL-8 levels and adhesions [56]. Furthermore, it seems from a 22-patient cohort that pleural fluid cytokine levels are also implicated in spontaneous pleurodesis in patients treated with indwelling pleural catheter for malignant pleural effusion, with IL-8, VEGF and bFGF (fibroblast growth factor), but not TGFb, being implicated in this procedure [57].

It has been long demonstrated that pleural exudates demonstrate increased procoagulant activity and depressed fibrinolytic activity leading to fibrin deposition in comparison to transudates (Figure 13, Figure 14 and Figure 15); early in vitro studies revealed that mesothelial cell themselves show procoagulant activity due to tissue factor that binds factor VII at the cell surface to initiate coagulation [58].

In addition, mesothelial cells contain tissue plasminogen activator (tPA) and urokinase plasminogen activator (uPA), but no fibrinolytic activity probably due to release of plasminogen activator inhibitors-1 and -2 (PAI-l and PAI-2) as well as antiplasmins [58]. The use of fibrinolytic agents (such as streptokinase, urokinase, tPA, PAI-1-neutralisisng antibodies, single-chain urokinase plasminogen activator), and DNAase are the basis of intrapleural fibrinolytic therapies aiming to disrupt the fibrine adhesions via activation of the plasminogen and to reduce the viscosity of the fluid through breakdown of extracellular DNA [20]. Intrapleural fibrinolytic therapy is associated with subsequent production of large effusions—this helps by may having a somewhat lavage effect [20]- probably mediated by the MCP-1 [59].

### 4.3. The Fibrinolytic Pathway

The importance of the coagulation/fibrinolysis balance in the pleural cavity has been studied in patients who underwent thoracoscopy due to benign or malignant pleural effusions, showing strong activation of coagulation and production of PAI in all patients including those without talc pleurodesis [60]. Fibrinolytic activity declined after talc poudrage in patients with successful pleurodesis, as opposed to those who failed pleurodesis and those without pleurodesis [60]. Interestingly, these pleural adhesions, as investigated in an animal model, seem to be well-formed structures resembling more to normal pleural tissue since they contain blood vessels, lymphatics and nerve fibers and are covered by mesothelial cells, than to a simple scar [61]. Fibrin is also important because it probably favors the implantation of cancer cells on injured tissues, as shown using injured peritoneal zones in a mouse model [62].

All these steps of pleural inflammation resemble the proposed mechanisms following pleurodesis, a condition that could serve as an example of pleura inflammation/fibrosis model and is nicely reviewed by Mierzejewski et al. [63]. Briefly, the intrapleural administration of sclerosing agents are believed to act mainly and directly on mesothelial cells which initiate an inflammation cascade by secreting various mediators such as IL-8 and TGFb; this will cause the influx of neutrophils and the activation of fibroblasts, respectively [63]. Neutrophils will further produce other mediators such as IL-1, IL-6, IL-8, IL-12 and MMPs [63]. The sclerosing agents will also cause an imbalance of the coagulation/fibrinolysis in the pleural space, since tPA (anticoagulant normally produced by mesothelial cells) will be inhibited by PAI-1, also produced by mesothelial cells after induction by TGFb; this will result in plasmin production inhibition and thus deposition of fibrin in the pleural cavity, which then be replaced by dense collagen fibers, ideally obliterating the pleural space [63].

### 4.4. Stem Cells and Tumour Growth

An interesting question of pleural pathophysiology is whether there are stem cells in these serosal cavities that could intervene in the repair process of inflammatory or tumoral processes. In addition, why some cases of pleural inflammation will resolve with re-establishment of the mesothelial layer, while other will repair with fibrosis, what influences the healing process? Regarding the last question, going back to 1984, Whitaker and Papadimitriou had already performed an animal study using the mesothelium of the testis as site of injury to investigate the mesothelial healing by using several approaches [64]. At that time, the principal theories proposed for this healing were: (a) mature mesothelial cells from nearby sites repopulate the injury, (b) free-floating serosal reserve cells settle on the injury and differentiate into mesothelial cells, (c) macrophages settle on the injury and differentiate into mesothelial cells, and (d) bone marrow cells give rise to the cells that will repair the injury [64].

The experiments performed by Whitaker and Papadimitriou showed that macrophages do initially cover he injury but actually it is the existing mesothelial cells that proliferate and move toward the wound center to cover the denuded surface in an initially immature form where they mature into mesothelial cells [64]. The repair process after mesothelial injury is more extensively studied in abdominal/pelvic operations, where the aim is to regenerate the mesothelium ideally without fibrotic adhesions between the two layers; however, this process is often imbalanced leading to postoperative adhesions [65]. One of the suggested research fields aiming to prevent this imbalance is tissue-engineering technologies which will try to reperitonealize denuded serosal surfaces [65]. In an animal model, autologous peritoneal grafts prevented adhesions, and the transplanted mesothelial cells should face the abdominal cavity (and not the underlying fibrous tissue) to achieve regeneration [65]. 

Similarly, the epicardium, the mesothelial tissue enveloping the heart is considered a multipotent cardiac progenitor tissue that contributes cells (smooth muscle cells, pericytes, fibroblast, adipocytes, endothelial cells, cardiomyocytes) and signals during heart development and regeneration, and identification of pro-regenerative subsets of epicardial cells is an active field of research [66]. This kind of studies in the pleura are largely lacking. A recent study reports for the first time, to the best of our knowledge, the isolation of pleural progenitor cells for tissue engineering purposes [67]. Notably, the pleura-derived cells showed chondrocyte-related and not osteoblast-related genes induction in comparison to pericardium-derived cells which showed the inverse genes induction [67]. It would be interesting to further investigate the presence, characteristic and roles of pleural mesothelial stem cells.

Into this direction, it is important to notice that the pleural fluid seems to have properties that help cell viability in vitro compared to serum-free and serum-enriched medium [68], and that cancer cells cultures proliferate similarly well in malignant and non-malignant pleural fluid without the addition of any other nutrients [69], suggesting that the pleural fluid itself contains elements that can sustain cancer cell proliferation. Despite these in vitro observations that could suggest that sustained pleural effusion can “help” cancer cell surviving, and that therefore pleural fluid exposure duration could be a negative prognostic factor, a recent retrospective study of 761 patients diagnosed with mesothelioma showed that the effusion’s duration is not associated with survival nor is the presence or absence of effusion during the initial diagnosis or the size of the initial effusion [70].

## 5. Non-Specific Pleuritis

When no granulomatous inflammation or tumor diagnosis is made in pleural biopsies, then a description of non-specific findings, such as inflammation and fibrosis, is usually provided by the pathologist. This accounts for the diagnosis of “non-specific pleuritis” (NSP), an observation made in almost 14% of patients with undiagnosed pleural effusion undergoing thoracoscopy [71]. The questions in these cases are: first, is this “benign” diagnosis a certain or instead a false-negative diagnosis hiding an undiagnosed malignancy, and second, could a more “specific diagnosis” be hypothesized by the histological findings? Starting from the last question, the only relative study trying to respond is, to the best of our knowledge, a study of 100 consecutive pleural biopsies with NSP [72]. The underlying etiology was in 28% of the cases pneumothorax, in 27% bacterial infection, in 10% cardiac etiologies, in 7% drug-induced, in 7% of viral etiology, and in 5% autoimmune [72]. The predominant inflammatory cell type was histiocytes often associated with eosinophils in pneumothorax, neutrophiles in bacterial causes, and lymphocytes in all other causes [72]. The most severe, and usually cellular and layered fibrosis, as well as a rich vascular proliferation was found with bacterial etiologies, while oedema did not significantly differ between etiologies [72].

Thus, an attempt to recognize the underlying etiology by morphological criteria could be made based in the following patterns: a severely fibrotic pleura with cellular layered fibrosis, vascular proliferation and neutrophils will most probably correspond to a bacterial cause, while a similarly impressive fibrosis but rich in lymphocytes could be associated with autoimmune disease [72]. Milder fibrosis is seen with viral and cardiac etiologies [72]. The prominent remodeling of the pleura in bacterial and autoimmune diseases likely reflects an imbalanced response in favor of the coagulation cascade, in comparison to other causes, while the reaction after pneumothorax does not evoke important tissue remodeling [72].

However, the most important question after a NSP diagnosis is whether a true malignancy is hidden, thus NSP being a false-negative diagnosis. In order to respond to this question, one should revise studies offering long-term follow up after NSP diagnosis (Table 2).

These studies revealed that a malignancy will be finally diagnosed in 0.5 to 15% NSPs. In most cases, the diagnosis of cancer—mesothelioma or metastasis—will be established shortly, usually few months after the initial thoracoscopy, probably corresponding to diagnostic errors. These can be related to the non-representative biopsy of the pleura obtained by the clinician or to a missed diagnosis by the pathologist. The first possibility is associated to the frequently complicated macroscopic appearance of the pleural cavity making it difficult to select the biopsy sites. A recent study describing the macroscopic features of the pleura during thoracoscopy showed that pleural disease of benign etiology presented principally an inflammatory macroscopic aspect with the most affected areas being the middle and inferior lateral costoparietal pleura, while in the malignant group, nodules were the predominant finding actually affecting the same localization [80].

However, while nodules were the predominant pattern in malignant pleural disease, they were also detected in 24% of benign etiologies [80]. In a study of 100 consecutive pleural biopsies with benign diagnosis (no cancer), the predominant macroscopic pattern was inflammatory in 40% of cases, fibrinous in 14%, septated in 8%, fibrous in 6%, hemorrhagic in 2%, and pleural plaques in 2% [72]. A severely fibrotic pleura could pose difficulties in obtaining deep biopsies to prove fat tissue invasion [79]. Regarding the second possibility of a pathological misdiagnosis, this could be attributed to lack of adipose tissue invasion, to paucicellular tumors, and lack of ancillary techniques [79,81].

However, there are some cases of NSP revealing malignancy, and most specifically mesothelioma, long after the initial thoracoscopy, which probably excludes a non-representative biopsy or a misdiagnosis, and suggests a long-lasting disease, which leads us to the case of pre-invasive mesothelial lesions. Does a precursor to mesothelioma exist? Until a few years ago, the main differential diagnosis of mesothelioma (apart from metastatic disease that could be usually easily diagnosed based on the clinical context and the immunoprofile of the lesion) was the so-called “atypical mesothelial hyperplasia” a term adopted by the WHO classification as a purely surface mesothelial proliferation that might or might not be malignant [82]. The most reliable criterion of malignancy has always been the invasion of the underlying adipose tissue, however, when this was not found at that time, diagnosis was becoming difficult, even between experts [83]. The follow-up of these patients revealed, not surprisingly, that almost half did harbor a malignant disease, further supporting the difficulty in correct diagnosis in some cases [83]. Thus, until that time, the “atypical mesothelial hyperplasia” probably corresponded to difficult to diagnose cases, and not necessarily to a truly pre-invasive disease. However, in the most recent years, new ancillary techniques became available [84], such as BAP1 loss in immunohistochemistry and *CDKN2A* homologous deletion identified by FISH or by its surrogate MTAP cytoplasmic immunohistochemical expression that allowed to diagnose mesothelial malignancy more confidently. Given the more reliable diagnosis achieved with the aid of these techniques, interest regarding the possibility of pre-invasive lesions to exist has been renewed. 

The mesothelioma in-situ has been already used as a term 30 years ago by Whitaker, Henderson and Shilkin, however, given the absence of reliable techniques at that time, “the mesothelioma in situ could be diagnosable only when invasive epithelial mesothelioma is demonstrable in the same specimen, in a follow-up biopsy, or at autopsy” [85,86]. The main concern with this definition is that the surface cells considered to represent the in-situ phase of the tumor could also correspond to a simple surface growth of the underlying mesothelioma [87], as seen for example with other neoplasias, such as the mammary Paget disease [88]. Then, two cases of mesothelial lesions characterized by only surface cells and no invasion (or minute invasive foci) but harboring the aforementioned molecular abnormalities were suggested as being true mesothelioma in situ cases [87]. Similarly, another case of proposed mesothelioma in situ [89] progressed to invasive disease almost five years later [90]. Following their initial observation, a group of experts in mesothelial pathology gathered another nine cases diagnosed with the following strict criteria: single layer of surface mesothelial cells with lost BAP1 expression, no evidence of invasive tumor by imaging techniques or during visual inspection of the serosal cavity, and no invasive mesothelioma diagnosed for at least 12 months after initial biopsy to avoid initial misdiagnoses [91]. In most cases, there were recurrent effusions [91]. Mesothelial cells were flat or cuboidal showing no or minimal atypia [91]. An invasive mesothelioma did develop in seven patients with a time interval of 12–92 months (median 60 months) [91]. After these observations, the mesothelioma in situ is for the first time included in the current WHO classification, and the essential criteria for its diagnosis are: non-resolving pleural effusion, no thoracoscopic or imaging evidence of tumor, single layer of mesothelial cells on the surface without invasive growth, and loss of BAP1 and/or MTAP and/or *CDKN2A* homozygous deletion, and multidisciplinary discussion of the diagnosis. Despite these guidelines, there still are inconsistencies in how the diagnosis is made even among experts, as shown in a recent survey of 34 pathologists [92]. Almost 70% of them had made or suggested the diagnosis of MIS in their practice; the diagnosis had been made between 0 and >20 times by individual pathologists, but the diagnosis was generally rare (two cases in a database of 4677 specimens and seven in a database of 3214 cases), and made the last two to four years, in comparison to the databases that spanned 40 years of practice [92]. Interestingly, only 9% denoted that accept the diagnosis only if the mesothelium is flat, whereas 65% accept flat or papillary mesothelium [92]. Another intriguing question in the case of mesothelioma in situ is if all cases will progress to invasive neoplasias. 

It is still unknown how to predict if and when this will happen [92]. The reported cases with subsequent invasion span a period of one [91] to 15 years [93] for the mesothelioma in situ to progress. Accepting that pre-invasive mesothelial lesions do exist could explain the long follow-up for diagnosing invasive mesothelioma after an initial negative biopsy in patients with recurrent pleural effusions [79]. It is interesting that patients with invasive mesotheliomas harboring an in-situ component have better prognosis than patients whose tumors do not show any in situ component [94]. Furthermore, some cases of mesothelioma in-situ could be missed if the diagnosis is relied only in BAP1 lost expression, since other molecular can also be implicated [94]. It is worth mentioning that these observations could pose obvious difficulties in the cytological diagnosis of mesothelial lesions [92,95].

Another question regarding NSP and cancer is whether chronic inflammation of the pleura can lead to malignancy. It is well-known that asbestos fibers induce mesothelial cell carcinogenesis through prolonged tissue damage, inflammation and repair, with various mechanisms being proposed as implicated in this carcinogenesis, including the induction of a pro-tumoral inflammation in the pleural cavity [82,96]. The high mobility group box 1 (HMGB1) protein released by mesothelial cells and macrophages after asbestos exposure promotes chronic inflammation [97] but also induces autophagy in mesothelial cells promoting their survival and malignant transformation [98]. Besides, carriers of the heterozygous germline *BAP1* mutation can be affected by any cancer type, but they are mostly susceptible to asbestos carcinogenesis and develop mesotheliomas, a “preference” suggested to be attributed to the higher amounts of HMGB1 secreted by these mutant cells [97]. 

Can non-asbestos-induced inflammation also promote mesothelial cell carcinogenesis? There is a substantial fraction of mesotheliomas that have no history of asbestos exposure, this being greater in the United States than in European countries, more in women than men, and greater in peritoneal than pleural mesotheliomas; almost 90% of peritoneal mesotheliomas in US women are likely unrelated to asbestos [99]. Other causes, such as different than asbestos mineral fibers, radiation exposure or a genetic predisposition can be involved; regarding chronic inflammation, only rare reports exist in cases of recurrent peritonitis or long-standing empyema associated with mesothelioma [99]. Peritoneal mesotheliomas have been reported in the context of endometriosis, and the chronic inflammation [100] seen in these cases has been elicited as a possible etiopathogenetic factor, however, other factors, such as personal/familial history of malignancies could also predispose these patients in developing mesothelial malignancy [101]. Similarly, in a database of 3800 mesothelioma patients, three (0.08%) had Crohn disease, while no ulcerative colitis cases were found, also suggesting a role for the transmural inflammation in this context [102]. In the context of peritoneal inflammation, we should also be aware of the mesothelial cysts, formerly named “cystic mesothelioma”, and the adenomatoid tumors, both mesothelial lesions potentially associated with inflammatory conditions [103,104], but usually of benign nature.

## 6. Conclusions

To conclude, the pleura is a cavity with fascinating pathophysiology ranging from simple fluid accumulation to tumors development, all inducing important consequences in patents health, and usually having an important association with local inflammation. Understanding the pathophysiology of pleural inflammation helps the development of the correct treatment strategies and opens new windows in pleural research.

## Figures and Tables

**Figure 1 cancers-14-01415-f001:**
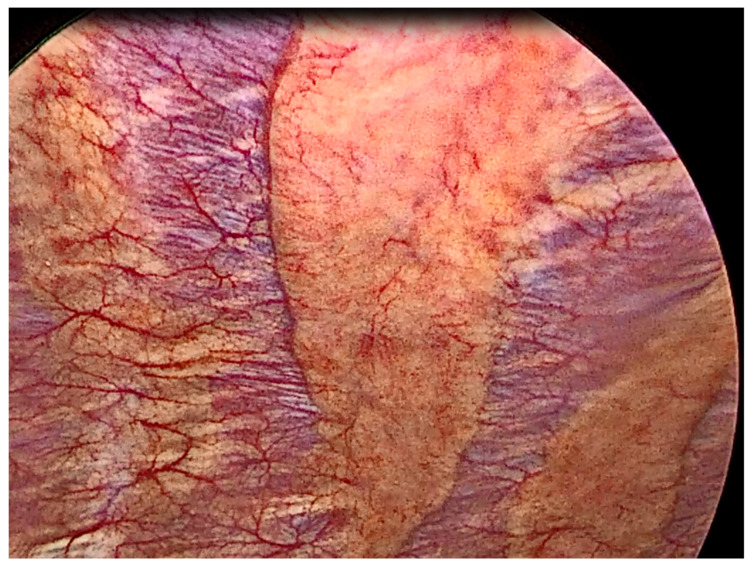
Thoracoscopic view of normal pleura.

**Figure 2 cancers-14-01415-f002:**
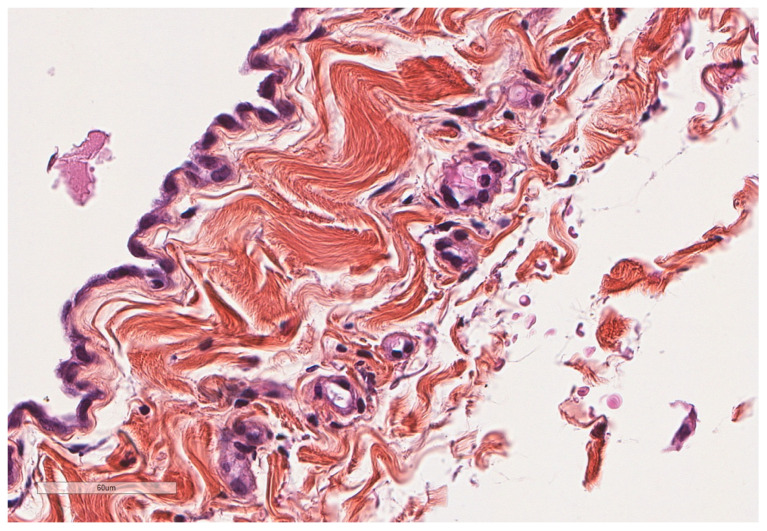
Normal parietal pleura microscopy. At the left side of the image, a single layer of mesothelial cells is seen underlined by a thin connective tissue. Hematoxylin, eosin, saffron staining (×400).

**Figure 3 cancers-14-01415-f003:**
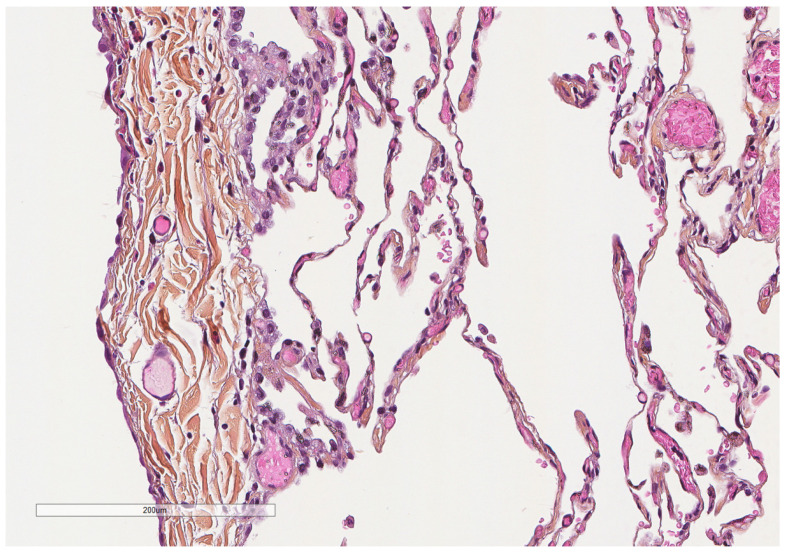
Microscopic aspect of the visceral pleura (left) covering the underlying alveoli. Hematoxylin, eosin, saffron staining (×200).

**Figure 4 cancers-14-01415-f004:**
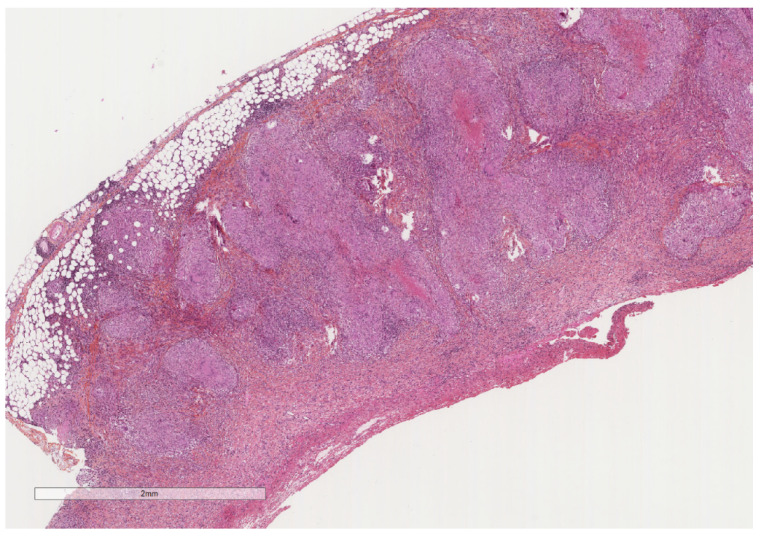
A severely thickened parietal pleura due to multiple nodules (see next figure for higher magnification). The adipose tissue is seen at the upper part of the image, while a fibrinous material covers the pleural cavity surface (lower part of the image). Hematoxylin, eosin, saffron staining (×20).

**Figure 5 cancers-14-01415-f005:**
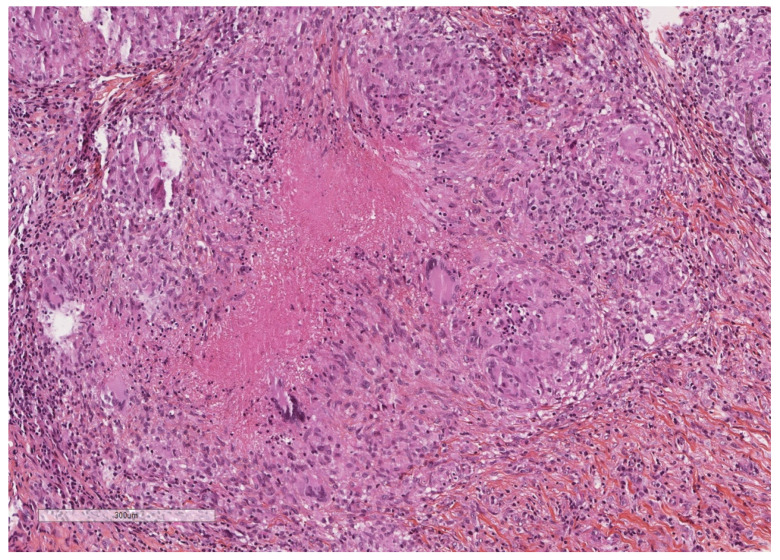
Epithelioid granulomas with giant cells and central necrosis characterize this pleural tuberculosis. Hematoxylin, eosin, saffron staining (×400).

**Figure 6 cancers-14-01415-f006:**
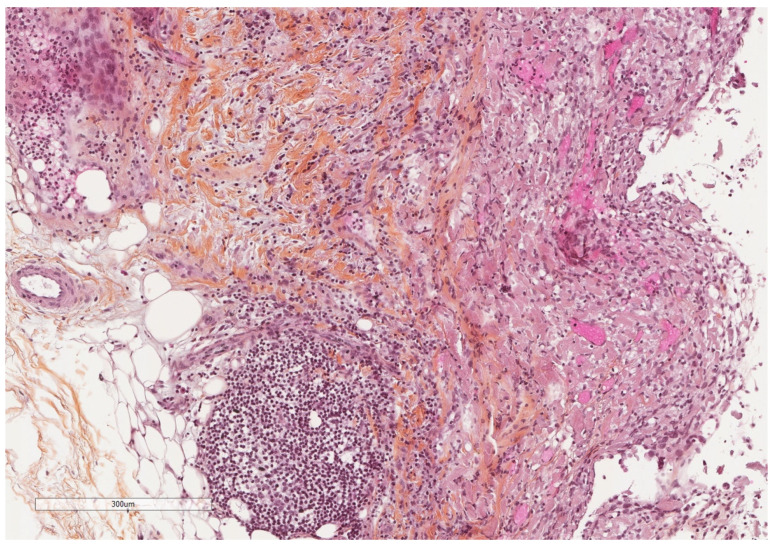
Microscopic image of a mildly thickened pleura (adipose tissue at the left, pleural cavity at the right) showing some chronic inflammation. Hematoxylin, eosin, saffron staining (×100).

**Figure 7 cancers-14-01415-f007:**
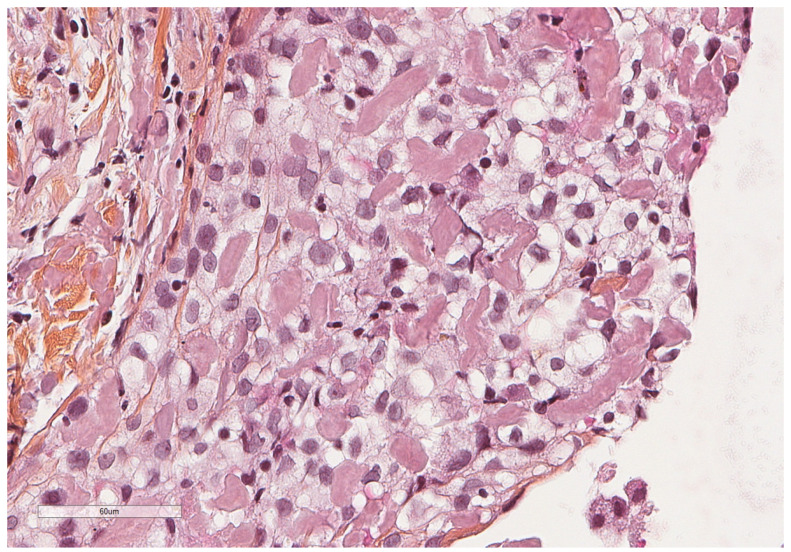
At higher magnification, an amorphous, eosinophilic, extracellular material is seen, corresponding to amyloid deposits. Hematoxylin, eosin, saffron staining (×400).

**Figure 8 cancers-14-01415-f008:**
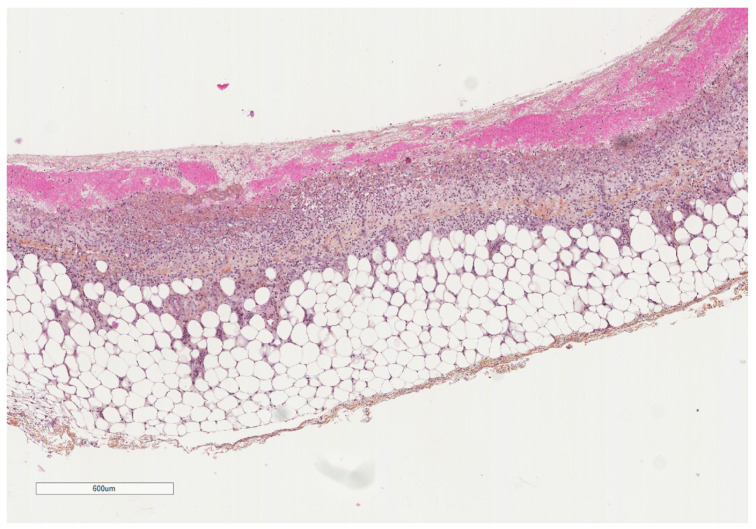
A parietal pleura biopsy showing a hemorrhagic layer in the pleural cavity (upper part) underlined by some inflammatory cells (see next figures for details); adipose tissue at the lower part. Hematoxylin, eosin, saffron staining (×40).

**Figure 9 cancers-14-01415-f009:**
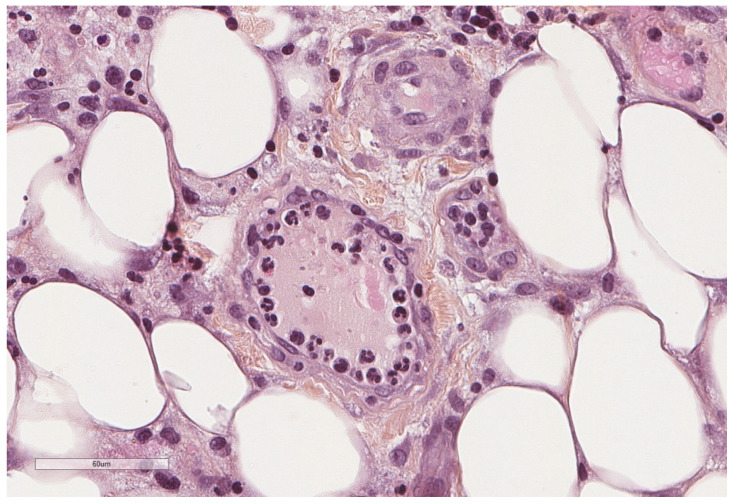
At this initial stage, margination, preceding diapedesis, of neutrophils inside blood vessels is seen. Hematoxylin, eosin, saffron staining (×400).

**Figure 10 cancers-14-01415-f010:**
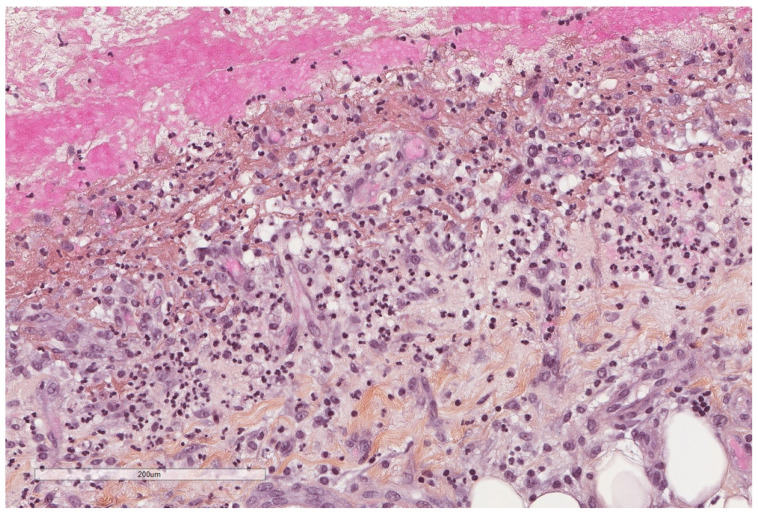
At this initial stage, neutrophils predominate. Hematoxylin, eosin, saffron staining (×400).

**Figure 11 cancers-14-01415-f011:**
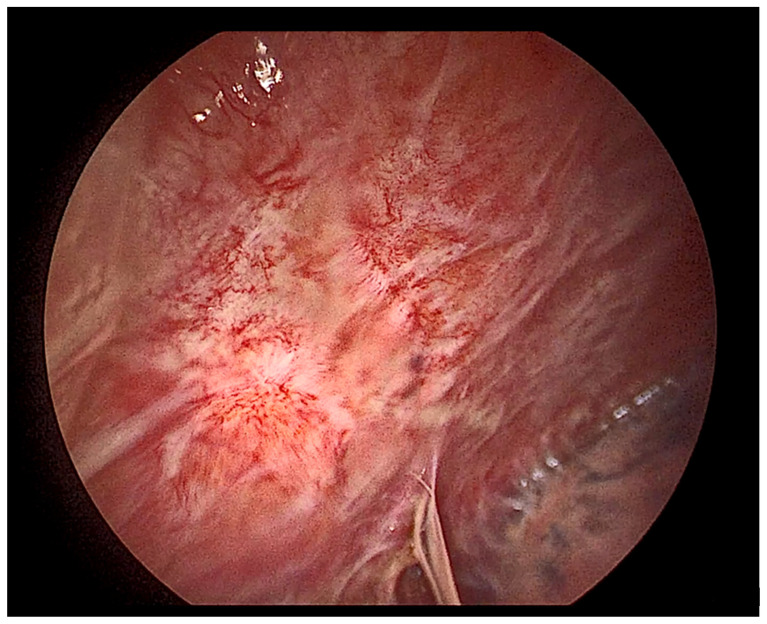
Thoracoscopic view of a mildly inflamed pleura.

**Figure 12 cancers-14-01415-f012:**
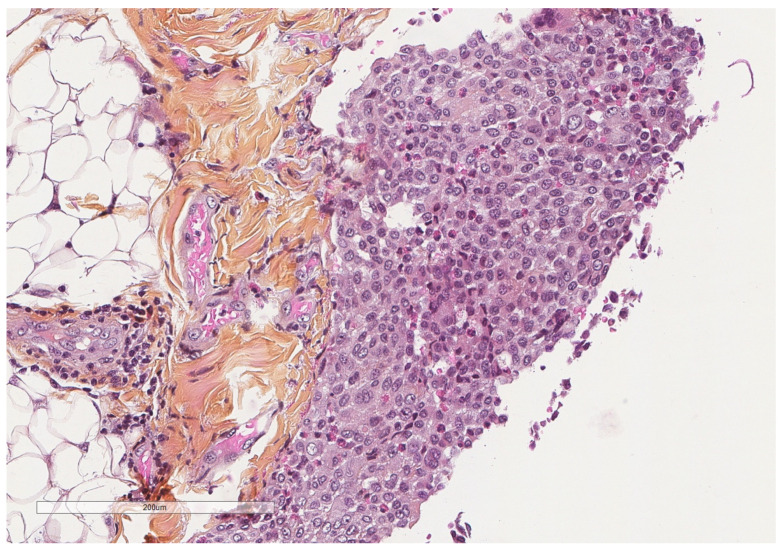
In pneumothorax-associated pleural effusion, macrophages intermixed with eosinophils predominate. Hematoxylin, eosin, saffron staining (×400).

**Figure 13 cancers-14-01415-f013:**
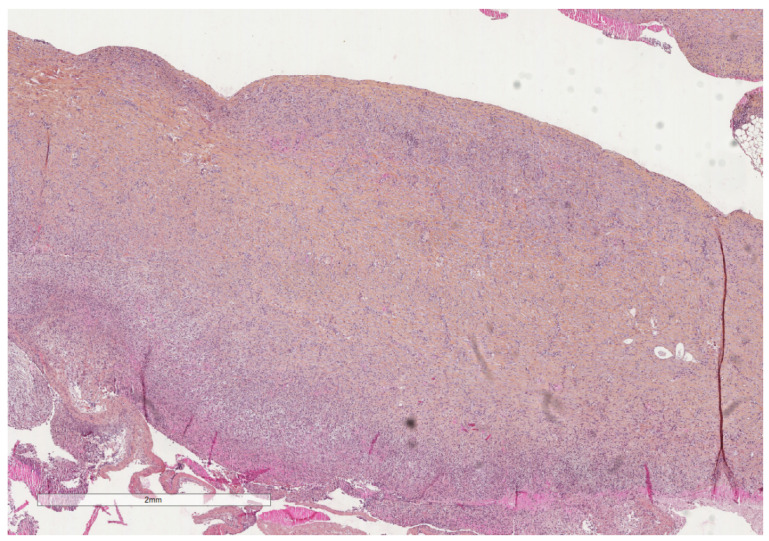
At this later stage, the pleura is severely thickened due to fibrosis (see next image for higher magnification). Hematoxylin, eosin, saffron staining (×20).

**Figure 14 cancers-14-01415-f014:**
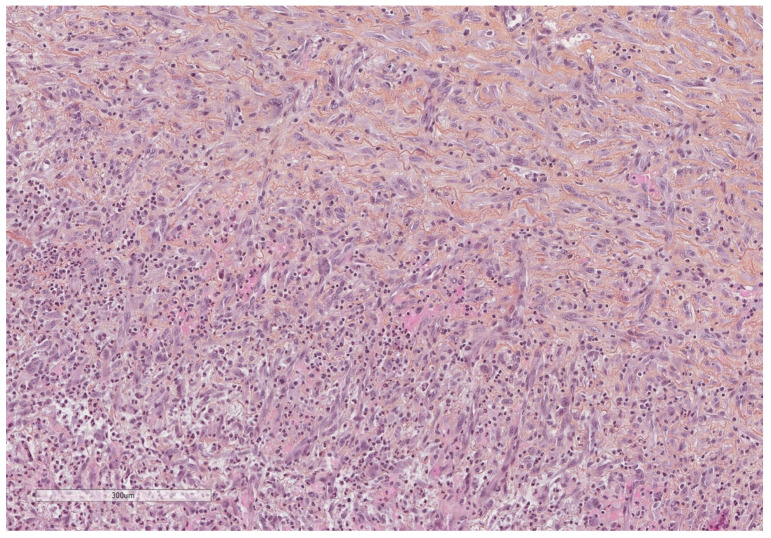
Neutrophils infiltrate and vascular proliferation seen towards the pleural cavity surface (lower part of the image), whereas dense collagenous tissue replaces this inflammation in the deepest parts of the pleura (upper part of the image). Hematoxylin, eosin, saffron staining (×400).

**Figure 15 cancers-14-01415-f015:**
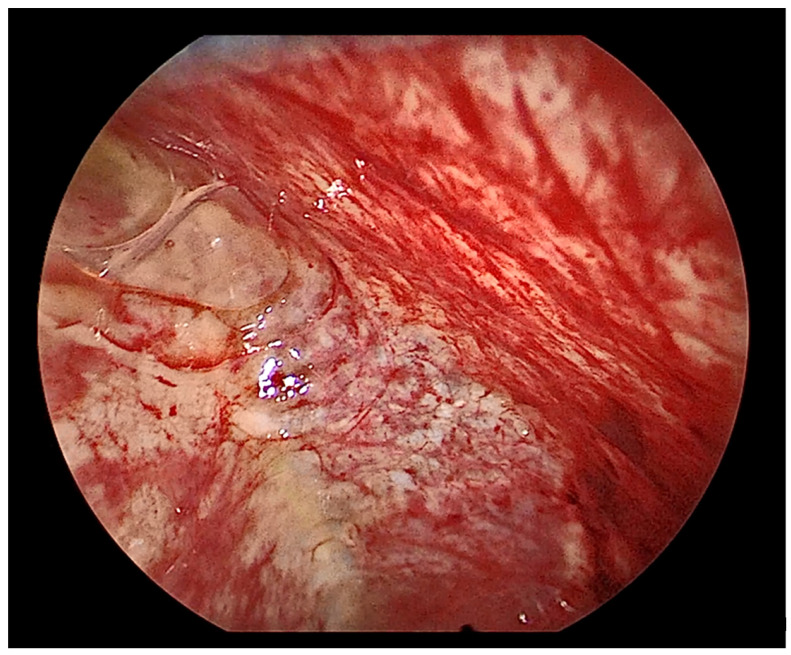
Thoracoscopic view of a severely inflamed pleura.

**Table 1 cancers-14-01415-t001:** Highlights on pleural effusion etiology.

Pleural Infection	6.7–9.9 cases/100,000 population Mortality of almost 20% at 12 months Principal pleural infections: *Staphylococcus aureus*, *viridans*-group *Streptococci*, *Streptococcus pneumoniae*, Enterobacteriaceae and Pseudomonas species Empyema shows the longer hospitalization stay and the higher cost between pleural effusion hospitalizations.
Abdominal Etiologies	5–15% of patients with cirrhosis suffer from hepatic hydrothorax Acute pancreatitis patients will present pleural effusion in half of the cases Chronic renal failure is associated with pleural effusion in almost 7% of patients
Connective Tissue Diseases	Incidence of pleural effusion is reported in 5–20% of rheumatoid arthritis patients, 17–60% of systemic lupus erythematosus patients, in 5–55% of granulomatosis with polyangiitis patients and in 7% of systemic sclerosis patients
IgG4-Related Disease	Thoracic involvement in almost half of the cases
Amyloidosis	Pleural effusion may be present in almost 40% of patients

For details see Section 3.

**Table 2 cancers-14-01415-t002:** Main findings of studies reporting on the follow up of non-specific pleuritis patients.

	Number of Patients	Follow Up	Malignancy
Venekamp 2003 [73]	60	32.9 ± (27.4) months	5 patients (8.3%): 3 mesothelioma 2 lung cancer
Davies 2010 [74]	44	21.3 ± 12 months	5 (12%) all mesothelioma after 9.8 ± 4.6 months
Depew 2014 [75]	86	1824 ± 1032 days	3 (3.5%) all mesothelioma after 205 ± 126 days
Gunluoglu 2015 [76]	53	24 months	2 (3.7%)
Yang [77]	52	35.5 ± 40.9 months	8 (15.4%)
Vakil 2017 [78]	172 patients with known malignancy: their pleural biopsies revealed malignancy in 42% of the cases, NSP in 52% and eosinophilic pleuritis in 5%		3 of the NSP group revealed malignancy
Karpathiou 2020 [79]	295 179 VATS 116 MT	47.3 ± 20.7 months	VATS: 1 (0.5%) at 5 years MT: 2 (1.7%) at 32 and 64 months All mesothelioma
Yu 2021 [71]	154 In 67 of the 154 patients with NSP (43.5%), no exact cause of their condition could be determined. They were eventually diagnosed with idiopathic pleural effusion.	61.5 ± 43.7 months	19 (12.3%): 7 lung cancer, 6 mesotheliomas, 2 gynecological tumors, 1 breast tumor, 1 prostatic cancer, 1 plasmacytoma, 1 thymoma at 3.3 ± 3 months

All retrospective studies. All NSP diagnosed after thoracoscopy. MT = medical thoracoscopy, VATS = video-assisted thoracoscopic surgery.

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
