# Peer review of "Inflammation of the Pleural Cavity: A Review on Pathogenesis, Diagnosis and Implications in Tumor Pathophysiology"

_cancers, 2022, doi:10.3390/cancers14061415_

Round 1

Reviewer 1 Report

The authors did a creditable job in writing this comprehensive review all around the pathophysiology of inflammation in the pleural cavity and pleural effusions. Thereby, also underlining the role of inflammation for tumor development. At this time point, this review also contains many current sources, making this work a nice recommendation for anyone familiar with pleural pathology. The authors themselves also contribute much to the sources, emphasizing their familiarity with the field.

This work should be considered for publication after minor additions.

  1. Please give some information on the prognostic value of pleural effusions in lung cancer and viral infections, especially COVID-19.

Author Response

We thank the reviewer for taking the time to review our manuscript and for his kind comments. We now added some prognostic information as requested (changes in green).

Reviewer 2 Report

The manuscript submitted by Karpathiou et al on pleural cavity described the etiology and pathophysiology of pleural inflammation and their association with malignancy. The review is very vast, well explained and covered most of the studies related to pleural effusion. However, I have some suggestions about this article.

Comments:

  1. In abstract, author should re-write the line no. 18-20 for better understanding.
  2. The pleural effusion etiology part is very detailed, I would suggest author should add either a diagram which summarize the pleural etiology or a table which summarize all the studies mentioned in that part.
  3. Author should add a brief paragraph about therapeutic implications or treatment strategies of pleural inflammation.
  4. Author should carefully proofread the manuscript for spelling mistakes.

Author Response

We thank the reviewer for taking the time to review our manuscript and for his kind comments. We modified the text accordingly(changes in blue).